# Early mobilisation and rehabilitation in the PICU: a UK survey

Jacqueline Y Thompson ®,[1] Julie C Menzies,[2] Joseph C Manning ®,[3,4] Jennifer McAnuff ®,[5,6] Emily Clare Brush,[1] Francesca Ryde,[1] Tim Rapley ®,[7] Nazima Pathan ®,[8] Stephen Brett,[9] David J Moore,[10] Michelle Geary,[11] Gillian A Colville ®,[12] Kevin P Morris,[13] Roger Charles Parslow,[14] Richard G Feltbower,[15] Sophie Lockley,[16] Fenella J Kirkham,[17] Rob J Forsyth ®,[18] Barnaby R Scholefield,[1] On behalf of the PERMIT Collaborators, Paediatric Critical Care Society Study-Group (PCCS-SG)

For numbered affiliations see end of article.

**Correspondence to**
Dr Barnaby R Scholefield; b.scholefield@bham.ac.uk

## ABSTRACT

**Objective** To understand the context and professional perspectives of delivering early rehabilitation and mobilisation (ERM) within UK paediatric intensive care units (PICUs).

**Design** A web-based survey administered from May 2019 to August 2019.

**Setting** UK PICUs.

**Participants** A total of 124 staff from 26 PICUs participated, including 22 (18%) doctors, 34 (27%) nurses, 28 (23%) physiotherapists, 19 (15%) occupational therapists and 21 (17%) were other professionals.

**Results** Key components of participants' definitions of ERM included tailored, multidisciplinary rehabilitation packages focused on promoting recovery. Multidisciplinary involvement in initiating ERM was commonly reported. Over half of respondents favoured delivering ERM after achieving physiological stability (n=69, 56%). All age groups were considered for ERM by relevant health professionals. However, responses differed concerning the timing of initiation. Interventions considered for ERM were more likely to be delivered to patients when PICU length of stay exceeded 28 days and among patients with acquired brain injury or severe developmental delay. The most commonly identified barriers were physiological instability (81%), limited staffing (79%), sedation requirement (73%), insufficient resources and equipment (69%), lack of recognition of patient readiness (67%), patient suitability (63%), inadequate training (61%) and inadequate funding (60%). Respondents ranked reduction in PICU length of stay (74%) and improvement in psychological outcomes (73%) as the most important benefits of ERM.

**Conclusion** ERM is gaining familiarity and endorsement in UK PICUs, but significant barriers to implementation due to limited resources and variation in content and delivery of ERM persist. A standardised protocol that sets out defined ERM interventions, along with implementation support to tackle modifiable barriers, is required to ensure the delivery of high-quality ERM.

## INTRODUCTION

In the UK, approximately 20 000 children are admitted to paediatric intensive care units (PICUs) yearly,[1] and although most recover,

### What is known about the subject?

⇒ Early rehabilitation and mobilisation (ERM) interventions are safe, feasible and effective within adult intensive care, but the evidence base in a paediatric setting is limited.
⇒ In critically ill adults, ERM delivery is tailored according to the patient's cardiovascular support requirements, levels of consciousness and tolerance levels.
⇒ Barriers to implementing ERM in paediatric intensive care units have been described in North America, but little is known about these within the UK NHS setting.

### What this study adds?

⇒ Despite positivity toward the concept of early rehabilitation and mobilisation (ERM), less than 20% of UK paediatric intensive care units (PICUs) currently have an established ERM protocol to define ERM content and practice.
⇒ ERM initiation and delivery is collaborative, but there is wide variability on which patients can receive ERM and when this should be initiated.
⇒ The provision of a standardised protocol that sets out safe and defined ERM activities along with implementation support would tackle modifiable barriers of intervention delivery.
⇒ The most important barriers to ERM delivery in PICUs within the NHS are financial resources and staffing, lack of protocols for patient selection and ERM guidelines.

some develop longer-term physical, psychological and cognitive impairment.[2] These significant morbidities have been termed post-intensive care syndrome in paediatrics.[3] Early rehabilitation and mobilisation (ERM) encompasses patient-tailored interventions, delivered within 7 days of admission, individually[4] or in a bundled package[5] to patients within intensive care settings. It is provided by health professionals from multiple disciplines

and care-givers and may promote physical (eg, functional activities)[4] and non-physical (eg, psychological and cognitive)[6] recovery.

Within adult ICUs, ERM has been demonstrated to be safe, feasible and cost effective.[7 8] It can shorten the length of ventilation, shorten the duration of intensive care and hospital stay with economic benefit, improve long-term physical functioning and return to independence and is recommended by National Institute for Health and Care Excellence.[8] While ERM has been reported to be safe and feasible in PICUs,[9–11] the patient population is different, and ERM is less well defined, leading to variability in practice. This variation in practice and components of interventions delivered has also been described in European PICUs.[12] At present, no national standardised care pathway for ERM exists within UK PICUs.

This study forms part of the National Institute for Health Research (NIHR) funded PERMIT study (NIHR HTA: 17/21/06) investigating ERM in children. This work was set out to understand reported practices and perceptions of ERM within the UK PICU context.

Our objectives were to:
► Explore how healthcare professionals describe and administer ERM using a qualitative approach.
► Identify and describe current ERM practice using a quantitative approach.
► Understand and quantify perceived barriers and facilitators of ERM, presenting findings using descriptive analysis.

## METHODS

A web-based survey (administered through www.smartsurvey.co.uk) was developed that included 25 questions (online supplemental material 1). A patient representative was involved in the design and development of the survey. A pilot survey was conducted among multidisciplinary health professionals (n=40) teams within two PICUs to assess acceptability and comprehensiveness. There were very few missing responses; therefore, no questions were removed, but five questions were rephrased to add clarity. Pilot responses were excluded from the main survey analysis.

A UK Paediatric Intensive Care Society Study-Group (PICS-SG) member of each UK PICU (n=29) was contacted via email and requested to identify and cascade to members of their local multidisciplinary team (including at least one physiotherapist, doctor and nurse) to complete the survey. Participating PICUs were sent a survey link to distribute between May 2019 and August 2019. Three follow-up reminders were sent at weekly intervals to non-responders.

Likert scales with the following categories (and associated scores)—always (5), very often (4), sometimes (3), seldom (2) and never (1)—were used to express the frequency of practice or level of agreement (using median and IQR of score). The ranking of perceived ERM benefits was calculated using the sum of ranked scores of respondents' top five important benefits (5

points for 1st place, reducing to 1 point for 5th place). Statistical analysis was performed using R V.x64 3.5.1 (R Foundation for Statistical Computing, Vienna, Austria).

Open-ended responses were analysed using a qualitative content analysis approach.[13] Two researchers independently familiarised themselves with the data and conducted open coding, using NVIVO software for data management. Codes were then discussed, summarised and organised.[14 15] In relevant sections of the paper, free-text quotes from respondents are reported to add context and clarity.[16]

### Patient and public involvement

The pilot survey received input from multidisciplinary health professionals who were members of the UK PICS-SG. We also recruited a patient representative as part of the study team to ensure that the perspectives, experiences and preferences of children admitted to PICU were incorporated into the design and development of the survey. The patient representative was a parent of a child who had been admitted to PICU. They both contributed to the nature of questions considered in the survey to ensure the study outcome would be relevant to parents and members of the PICS-SG on dissemination.

## RESULTS
### Demographics

We received responses from PICS-SG link members in 26/29 (90%) UK PICUs. A total of 191 healthcare professionals opened the survey link, with 124 (65%) submitting responses.

**Table 1** Characteristics of survey respondents (n=124 respondents)

| Professional group | n (%)* |
|---|---|
| Nurse | 34 (27) |
| Physiotherapist | 28 (23) |
| Medical doctor (consultant) | 22 (18) |
| Occupational therapist | 19 (15) |
| Play therapist | 7 (6) |
| Psychologist | 7 (6) |
| Dietician | 6 (5) |
| Speech and language therapist | 1 (1) |
| **Years of experience** | **n (%)*** |
| <1 year | 7 (6) |
| 1 year to <5 years | 27 (22) |
| 5 years to <10 years | 30 (24) |
| 10 years to <15 years | 14 (11) |
| 15 years to <20 years | 33 (27) |
| More than 20 years | 15 (12) |

*Percentages may not total 100 due to rounding.

**Table 2** Current views of ERM in PICU (n=121 respondents)

| Current view of ERM in PICU | n (%)* |
|---|---|
| Crucial, should be the top priority in the care of PICU patients | 15 (12) |
| Very important, should be a priority in the care of PICU patients | 67 (55) |
| Important, should be a priority in the care of PICU patients | 35 (29) |
| Somewhat important, should be considered in the care of PICU patients | 4 (3) |
| Not of great importance, clinicians should bear it in mind in the care of PICU patients | 0 (0) |
| Of minimal importance to the care of PICU patients | 0 (0) |
| Of no importance to the care of the PICU patients | 0 (0) |

*Percentages may not total 100 due to rounding.
ERM, early rehabilitation and mobilisation; PICU, paediatric intensive care unit.

As shown in table 1, the majority of respondents were nurses (n=34, 27%), physiotherapists (n=28, 23%) and doctors (n=22, 18%). There were also responses from occupational therapists (n=19, 15%), play therapists (n=7, 6%), psychologists (n=7, 6%), dieticians (n=6, 5%) and speech and language therapists (n=1, 1%). Almost three-quarters of health professionals had ≥5-year experience, with 48 (39%) having ≥15-year experience. The majority of respondents considered ERM to be a priority, either crucial (15, 12%), very important (67, 55%) or important (35, 29%) in the care of PICU patients (table 2).

### Description of ERM
We invited respondents to describe ERM on their terms. Descriptions were provided by 104 (84%), which were summarised into four categories, 'activity focused', 'tailored', 'promote recovery' and 'timing of ERM' (see online supplemental file 2 and table 1). Overall, ERM was considered to be an individualised package of graded interventions based on an activity-focused programme, to reduce the sequelae of critical illness or injury. However,

responses differed for when ERM should be initiated, often emphasising the need for individualisation.

### Availability of established ERM protocols
Respondents were asked to describe the content of established ERM protocols within their PICU. Only 12 participants (10%) reported working in a PICU with an established ERM protocol (n=5/26, 19% of PICUs). The most common components of ERM protocols were 'physical therapy not requiring additional equipment' (n=9/12, 75%) and 'occupational therapy interventions' (n=8/12, 67%). Only 4/12 (33%) referred to play therapy or speech and language therapy, and no ERM protocol specified input from psychologists or psychiatrists. All participants were asked about the content of non-ERM protocols in their PICU. Only 18/124 (15%) reported that guidance for physical or occupational therapy activities existed in other non-ERM protocols within PICU (table 3).

### Recipients of ERM
Fifty-one (41%) respondents reported that all PICU patients 'always' or 'very often' received ERM (online supplemental file 2 and table 2). Overall, 14 (11%) respondents reported 'seldom' or 'never' delivering ERM.

ERM was reported to be more likely to be delivered to patients when PICU length of stay exceeded 28 days. Patients admitted for 28 days or more were more likely (n=91, 75%) to 'always' or 'very often' receive ERM compared with those admitted for shorter periods. Only 17 (13%) of those staying for less than 3 days, 44 (36%) of those admitted between three to 7 days and 73 (59%) of those admitted between seven to 28 days were more likely to receive ERM. Participants reported that patients with acquired brain injury (n=75, 60%) and severe developmental delay (n=54, 44%) were 'always' or 'very often' likely to receive ERM.

### Perceived benefits of ERM
Participants ranked the 5 most important potential benefits of ERM out of 13 options (figure 1 and table 4). The most important outcomes identified were (1) reduced PICU length of stay, (2) improved psychological outcomes

**Table 3** Content of ERM and non-ERM protocols

| Items | Within an ERM protocol (n=12 respondents) Yes n (%) | Within a non-ERM protocols (n=124 respondents) Yes n (%) |
|---|---|---|
| Physical therapy requiring additional equipment | 9 (75) | 18 (15) |
| Occupational therapy interventions | 9 (75) | 18 (15) |
| Physical therapy not requiring additional equipment | 8 (67) | 17 (14) |
| Speech and language therapy interventions | 4 (33) | 12 (10) |
| Psychology interventions | 0 (0) | 8 (6) |
| Delirium screening | 0 (0) | 1 (1) |

ERM, early rehabilitation and mobilisation.

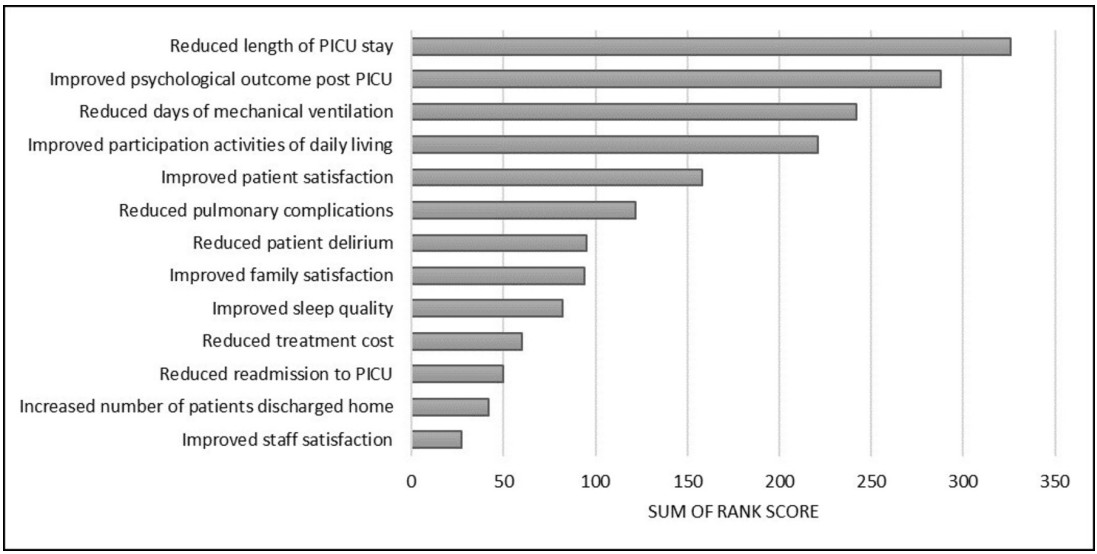

**Figure 1** Perceived benefits of ERM: ranking of participants' potential top 5 perceived benefits of delivering ERM within PICUs. Sum of rank score: ranking of top 5 (1–5) (1st placed rank scored 5 points to 5th placed scored 1 point). 121/124 (98%) participants ranked scores. ERM, early rehabilitation and mobilisation; PICU, paediatric intensive care unit.

for patients after PICU, (3) reduced days of mechanical ventilation, (4) improved participation in activities of daily living and (5) improved patient satisfaction.

**Table 4** Top 5 most important benefits of ERM

| Perceived ERM benefits (most to least important) | Sum rank score* |
|---|---|
| Reduction in length of ICU stay | 326 |
| Improvement in the psychological impact of PICU care | 288 |
| Reduction in days requiring mechanical ventilation (MV) | 242 |
| Improvement in daily life participation following discharge | 221 |
| Improved patient satisfaction | 158 |
| Reduction in the rate of pulmonary complications | 122 |
| Reduction in patient delirium | 95 |
| Improved family satisfaction | 94 |
| Improvement in patient sleep quality | 82 |
| Reduction in treatment cost | 60 |
| Reduction in readmission | 50 |
| Increase in the number of patients discharged home | 42 |
| Improved staff satisfaction | 27 |

*Sum of rank score: ranking of top 5 (1–5) (1st placed rank scored 5 points to 5th placed scored 1 point). 121/124 (98%) participants ranked scores.
ERM, early rehabilitation and mobilisation; ICU, intensive care unit; PICU, paediatric intensive care unit.

### Initiation and delivery of ERM

The decision for ERM initiation was perceived by 96 (77%) to be primarily led by physiotherapists, 92 doctors (74%) and 64 bedside nurses (52%). Parents were felt to initiate ERM by only 24 of respondents (19%) (table 5).

The most influential factor in ERM initiation was reported to be patient stability (n=69, 56%). Other influential factors were the length of stay; 15 (12%) reported ERM was initiated within 24 hours, and 16 (13%) within 2–3 days of PICU admission. Only 5 (4%) of respondents would not consider ERM at all on PICU. The influence of perceived clinical stability is demonstrated in respondents' free-text comments:

> We are involved as early as required depending on the child/young person medical stability and their rehabilitation needs. (Occupational Therapist, 033)

**Table 5** Which professional or parent groups in PICU initiates ERM (n=124 respondents)

| Professional or family group | Yes n (%) |
|---|---|
| Physiotherapists | 96 (77) |
| Physicians | 92 (74) |
| Bedside nurses | 64 (52) |
| Senior nurses | 58 (47) |
| Other members of the medical team | 55 (44) |
| Occupational therapists | 37 (30) |
| Parents or family members | 24 (19) |

ERM, early rehabilitation and mobilisation; PICU, paediatric intensive care unit.

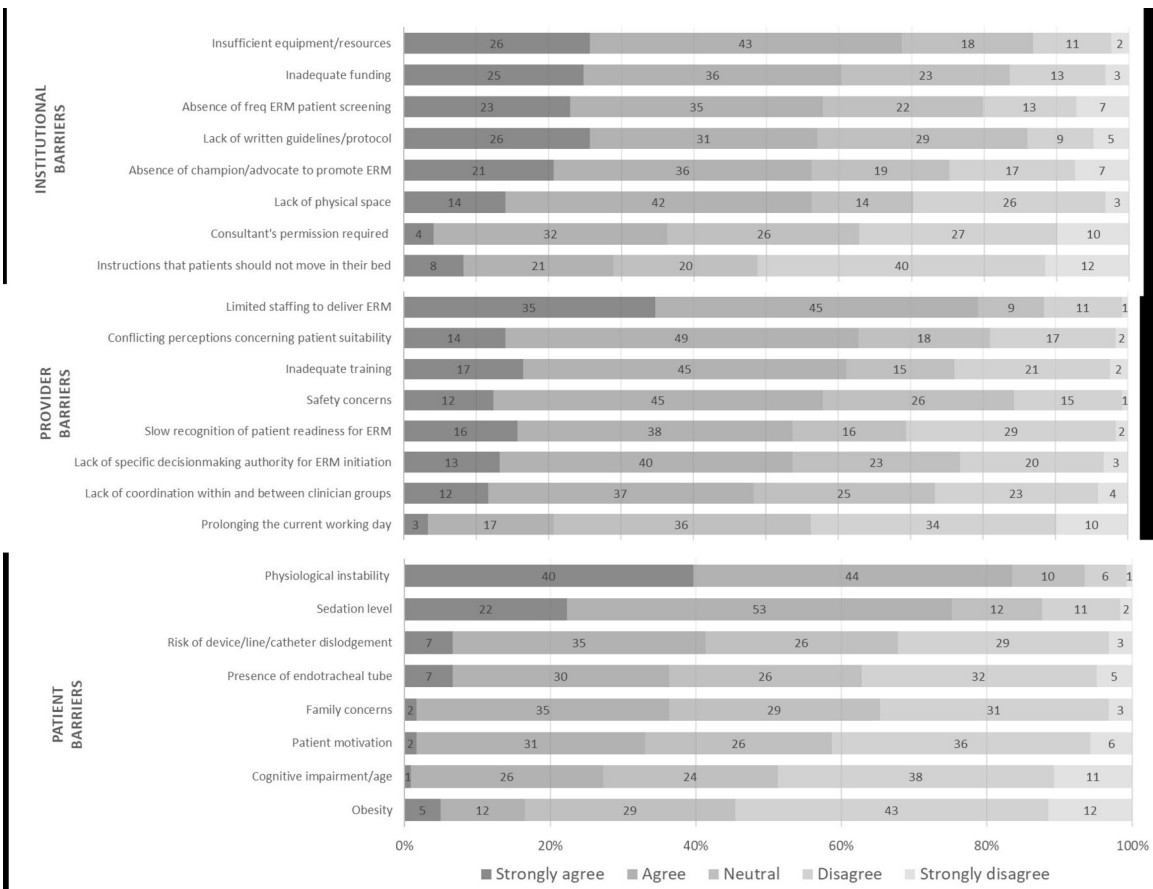

**Figure 2** Perceived barriers of ERM: institutional, patients and provider barriers to ERM. The percentage of responses for categories strongly agree, agree, neutral, disagree and strongly disagree shown. Responses ranked on the cumulative score of percentage 'strongly agree and agree'. ERM, early rehabilitation and mobilisation.

Usually ERM activity is not considered until patients can physiologically tolerate movement and are cardiovascularly stable. (Nurse, 008)

Assessment of patient stability and tolerance of ERM was less well described. Most respondents (n=98, 79%) provided subjective cues or informal clinical criteria. These included monitoring vital signs, physiological changes, observing behavioural changes and documenting adverse events.

Physiotherapists (n=113, 92%), nurses (n=103, 84%) and parents or family members (n=92, 75%) were 'always' or 'very often' involved in the ongoing delivery of ERM with less frequent input from other members of the multidisciplinary teams (online supplemental material 2 and table 2).

**Barriers to ERM implementation**

Figure 2 presents the perceived barriers of ERM (see also online supplemental material 2 and table 3). The most significant barriers identified as institutional barriers were insufficient resources and equipment ('strongly agree' or 'agree': 83, 69%) and inadequate funding (73, 61%). Participants provided examples of resources having to be shared across organisations or specially ordered to deliver ERM to patients.

All equipment shared with the whole therapy department at present, therefore, dependent on availability. (OT, 010)

A lack of established protocols (n=69, 57%), ERM champions (n=68, 57%), space (n=68, 56%) and robust patient screening processes (n=63, 58%) were also issues identified by respondents. Most PICUs had access to standard lifting 22/26 (85%) and specialist static seating equipment 25/26 (96%); however, bedside or in-bed cycling machines were only available in 10 (38%) of PICUs (see table 6).

Limited staffing was the most frequently reported barrier to providers delivering ERM, with 101 (79%) 'agreeing' or 'strongly agreeing'. Approximately half of the respondents agreed that issues such as training, patient safety, lack of decision-making authority and delays in recognising patients' ERM needs were barriers to ERM initiation. However, only 25 (21%) 'agreed' or 'strongly agreed' that the impact of ERM potentially prolonging the working day was a barrier.

The two most frequently reported barriers to delivering ERM at the patient level were physiological instability (n=101, 81% agreeing or strongly agreeing) and sedation (n=91, 73%). Over 50% (n=64) 'disagreed' or 'strongly disagreed' that obesity was a barrier.

**Table 6** Types of ERM equipment available in each PICU (n=26)

| ERM equipment available in each PICU | n (%) |
|---|---|
| Specialist static seating | 25 (96) |
| Portable ventilators | 23 (88) |
| Mobile lifts | 22 (85) |
| Tilt table | 22 (85) |
| Bed with full chair position | 18 (69) |
| Specialist wheelchair | 18 (69) |
| Bed with Trendelenburg features | 13 (50) |
| Patient rolling walker | 11 (42) |
| Bedside cycle or in-bed cycle | 10 (38) |
| Ceiling lifts | 8 (31) |
| Specialty bed with continuous side to side rotation | 8 (31) |
| Bed with retractable footboard | 7 (27) |
| Bed with chair egress exit out the foot of the bed | 5 (19) |
| Transcutaneous electrical nerve stimulation | 3 (12) |

ERM, early rehabilitation and mobilisation; PICU, paediatric intensive care unit.

## DISCUSSION

This national survey of healthcare practitioners from UK PICUs identified the importance of ERM as an intervention that participants believe can improve the physical and psychological recovery of critically ill or injured infants and children across all ages. Our findings indicate support for ERM but highlight uncertainty with suitability, variability with the definition of this complex intervention, variation in the timing of initiation and which patient groups should receive ERM. Key barriers to ERM delivery were identified (eg, funding and staffing) and potential clinical (eg, improved psychological outcomes) and economic (eg, reduced PICU length of stay) benefits to patients and PICUs were also identified.

How early is 'early' in ERM has been challenging to define for healthcare professionals? A time-based definition (eg, within the first 2–5 days of ICU admission) has been proposed[17]; however, this can conflict with the patient stability approach (eg, start as early as the patient is able to receive it). Our results indicate uncertainty and wide variation in time to start ERM (24 hours to over 7 days), increasing agreement for ERM to be considered after longer periods on PICU, and support for the concept of 'as early as the patient's clinical condition allows', which may be much longer. In the UK, only 43% of patients remain in PICU for 3 days or more.[1] The balance of delivering a programme of ERM to a large number of patients who may only receive intervention within a very short period versus targeting patients at a higher risk of prolonged PICU stay needs to be considered. However, it has been proposed that for ERM to become embedded in clinical practice, it should become a standard of care within 48 hours of ICU admission.[18]

The uncertainty of the content of ERM also adds to the challenge for healthcare professionals to appreciate when ERM could be delivered. Understandably, routine bedside nursing care (eg, functional positioning) may be considered acceptable earlier than more advanced physical therapies requiring multiple staff (eg, sitting a ventilated child out of bed or in-bed cycling). Our survey identified clinical stability as the most influential patient factor for initiation. The reported lack of ERM protocols in most (21/26) UK PICUs reinforces a strong requirement for evidence-based standardised protocols with optimal timing, intensity, frequency and duration of ERM. There is a need for flexible protocols to allow for tailoring rather than prescription.

ERM was more likely to be delivered to patients admitted for greater than 28 days among patients with acquired brain injury or severe developmental delay across all age ranges. This reflects the cross-over between ERM and established rehabilitation programmes following acquired brain injury[19] or for patients admitted to PICU with a pre-existing rehabilitation package. PICUs may be able to build on the success of established programmes to implement ERM to a wider critical care population and to use the existing multidisciplinary expertise, and evolving rehabilitation evidence base to support the adoption of effective treatments. To date, most published ERM intervention studies have excluded patients who were less than 3 years of age.[4 6] However, this represents 60% of the UK PICU patient population,[1] and this age group was as likely to receive ERM as older children in our study. Future ERM trials should include all PICU age groups to ensure ERM content and efficacy is assessed across all potential patients.

Our results show that doctors, physiotherapists and nurses have an equally important role in the decision to initiate ERM within the UK NHS setting. This contrasts with other countries and healthcare settings where doctor-led approval was required by 70%–81% of respondents, which was a potential barrier to mobilisation.[20 21] Nurses' and parents' roles are also important both in the initiation and delivery of ERM. In settings where nurses reported a low level (39%) of support for ERM, lack of involvement and understanding were key features.[22] In our study, 91% felt 'involved' in the delivery of ERM. However, healthcare professionals reported that parents were the least likely group to initiate ERM (19%), although becoming influential in its ongoing delivery. This is consistent with recent European[23] and North American[11 24 25] point prevalence studies in PICUs, highlighting the important role parents play in delivering ERM. Building on both the multidisciplinary support for ERM and empowering parents to initiate ERM may be potential strategies to improve implementation.

The key barriers to ERM practice were (1) at institutional level: insufficient resources, equipment and funding, (2) at provider level: limited staffing, training,

protocols and slow recognition of readiness for ERM and (3) at patient level: physiological instability, risk of endotracheal tube dislodgement and amount of sedation. These barriers have been previously described in other health settings[20–22 26] with proposed facilitators, including adopting formal protocol/manual[27] (a key feature lacking across most UK PICUs), local champions[27] and team engagement/collaboration.[27 28] Implementation of ERM within the NHS will require building on the established multidisciplinary teams, incorporating ERM within established protocols for weaning off mechanical ventilation and sedation and realistic goal setting. The potential benefits of ERM in terms of PICU length of stay and improvement in the psychological outcome for patients should be assessed within the core outcomes evaluating the efficacy of ERM within PICU.

The strength of this survey was an inclusive representation of 90% of UK PICUs and views from the wider multidisciplinary team. However, none or partial responses may indicate poor engagement in ERM topic, and as with all self-reported surveys, responses indicate reported rather than actual clinical practice. A limitation was the use of a non-validated questionnaire. We did not conduct questionnaire validation because it would be very difficult to standardise these questions within the time constraints of the study and the NIHR HTA funding. The study team considered the questionnaire suitable for the study aims after receiving two rounds of expert reviews. Finally, the findings represent the views of UK NHS staff and may not be generalisable to other healthcare settings.

## CONCLUSION

ERM, in some form, is currently delivered to critically ill or injured patients of all ages across UK PICUs, but significant barriers to full implementation exist due to resource limitations and lack of institutional and national guidance. The UK should build on the existing strong multidisciplinary support for ERM in PICUs. A standardised protocol that sets out defined ERM interventions and implementation support to tackle modifiable barriers is required to ensure the delivery of high-quality ERM.

**Author affiliations**
[1]Institute of Inflammation and Ageing, University of Birmingham, Birmingham, UK
[2]Department of Paediatric Intensive Care, Birmingham Women's and Children's NHS Foundation Trust, Birmingham, UK
[3]Nottingham Children's Hospital, Nottingham University Hospitals NHS Trust, Nottingham, UK
[4]Children and Young People Health Research, School of Health Sciences, University of Nottingham, Nottingham, UK
[5]Population Health Sciences Institute, Newcastle University, Newcastle upon Tyne, UK
[6]Department of Social Work, Education and Community Wellbeing, Northumbria University, Newcastle upon Tyne, UK
[7]Northumbria University, Newcastle upon Tyne, UK
[8]Department of Paediatrics, University of Cambridge, Cambridge, UK
[9]Department of Surgery and Cancer, Imperial College of Science, Technology and Medicine, London, UK
[10]Institute of Applied Health Research, University of Birmingham, Edgbaston, Birmingham, England
[11]Department of Child Health, University Hospital Southampton NHS Foundation Trust, Southampton, UK
[12]Paediatric Psychology Service, St George's University Hospitals NHS Foundation Trust, London, UK
[13]Birmingham Women's & Children's NHS Foundation Trust, Birmingham, UK
[14]Paediatric Epidemiology Group, University of Leeds, Leeds, UK
[15]University of Leeds, Leeds, UK
[16]Birmingham Children's Hospital, Birmingham, UK
[17]Neurosciences Unit, UCL Institute of Child Health, London, UK
[18]Institute of Neuroscience, Newcastle University, Newcastle upon Tyne, UK

**Acknowledgements** We thank the Paediatric Intensive Care Society Study Group, all members of the Paediatric Intensive Care Units and principal investigators for their contribution to the development of the PERMIT survey.

**Collaborators** Addenbrooke's Cambridge, Dr Nazima Pathan. Alder Hey Children's NHS Foundation Trust, Liverpool, Dr Petr Jirasek. Birmingham Women's and Children's Hospital NHS Foundation Trust, Dr Barnaby Scholefield. Bristol Royal Hospital for Children, Kate Baptiste. Evelina London Children's Hospital, Dr Miriam Fine-Goulden. Freeman Hospital, Newcastle upon Tyne Hospitals NHS Foundation Trust, Dr Deborah Cross. GOSH CICU, Great Ormond Street Hospital for Children NHS Foundation Trust, Emma Shkurka and Dr Peter Sidgwick. Great North Children's Hospital, Newcastle, Dr Rachel Agbeko. Kings College Hospital NHS Foundation Trust, Dr Bogdana Zoica. Leeds Teaching Hospitals NHS Trust, St. James's University Hospital, Dr Sian Cooper. Noah's Ark Children's Hospital for Wales, Dr Siva Oruganti. Nottingham Children's Hospital, Queen's Medical Centre, Dr Georgina Harlow. Oxford University Hospitals, Dr Deirdre O'Shea. Royal Belfast Hospital for Sick Children, located in Royal Victoria Hospital, Dr Julie Richardson. Royal Hospital for Children Glasgow, Dr Richard Levin. Royal Hospital for Sick Children in Edinburgh, Dr Tsz Yan and Dr Milly Lo. Royal Manchester Children's Hospital, Dr Ravishankar Nagaraj. Sheffield Children's Hospital NHS Foundation Trust, Dr Rum Thomas. Southampton Children's Hospital, Katy Morton. St Georges Hospital, London, Dr Soumendu Manna. St Mary's Hospital, Praed Street, Imperial London, Dr Tom Bycroft. University Hospital Leicester and Glenfield Hospital, Dr Sanjiv Nichani. University Hospital North Midlands, Dr Kanaris Constantinos. South Tees Hospitals NHS Foundation Trust.

**Contributors** BRS, JCMenzies, JCManning, FR, TR, NP, SB, JM, DJM, MG, GAC, KPM, RCP, RGF, SL and FJK developed the original study protocol. BRS, ECB, JCManning and FR developed the pilot survey and preliminary analysis. JYT, JCMenzies, JCManning and BRS developed the final survey, coordinated administration and interpretation and analysis of data. BRS and JYT drafted the manuscript, which has had input from all authors. All authors revised and approved the final manuscript. Paediatric Intensive Care Society Study-Group members reviewed the protocol and endorsed the survey. BRS the study guarantor, accepts full responsibility for the work and/or the conduct of the study, had access to the data, and controlled the decision to publish.

**Funding** This study/project is funded by the National Institute for Health Research (Health Technology Assessment Programme; project reference: 17/21).

**Disclaimer** The views expressed are those of the author(s) and not necessarily those of the NIHR or the Department of Health and Social Care.

**Competing interests** JCMenzies is funded as a National Institute for Health Research (NIHR) 70@70 Senior Nurse and Midwife Research Leader. JCManning is an NIHR Health Education England funded as an ICA clinical lecturer. BS receives funding as an NIHR Clinician Scientist Fellow, and RJF and TR receives NIHR grant funding.

**Patient and public involvement** Patients and/or the public were involved in the design, or conduct, or reporting, or dissemination plans of this research. Refer to the Methods section for further details.

**Patient consent for publication** Not applicable.

**Ethics approval** This study involves human participants. The University of Birmingham granted institutional ethical approval on 5 February 2019 (reference number: ERN_18–1134). Consent was implied through survey completion.

**Provenance and peer review** Not commissioned; externally peer reviewed.

**ORCID iDs**
Jacqueline Y Thompson http://orcid.org/0000-0002-9775-361X
Joseph C Manning http://orcid.org/0000-0002-6077-4169
Jennifer McAnuff http://orcid.org/0000-0002-1636-0049
Tim Rapley http://orcid.org/0000-0003-4836-4279
Nazima Pathan http://orcid.org/0000-0001-7656-9453
Gillian A Colville http://orcid.org/0000-0001-8530-2822
Rob J Forsyth http://orcid.org/0000-0002-5657-4180

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
