## [Reviewer comments · BMJ Paediatrics Open]

ARTICLE DETAILS

TITLE (PROVISIONAL)	Early Mobilisation and Rehabilitation in the PICU: A UK survey
AUTHORS	Thompson, Jacqueline Menzies, Julie Manning, Joseph; McAnuff, Jennifer Brush, Emily; Ryde, Francesca Rapley, Tim Pathan, Nazima Brett, Stephen Moore, David Geary, Michelle Colville, Gillian Morris, Kevin PARSLOW, ROGER Feltbower, Richard Lockley, Sophie Kirkham, Fenella Forsyth, Rob Scholefield, Barnaby

VERSION 1 – REVIEW

REVIEWER	Reviewer name: Dr. Ericka Fink Institution and Country: not applicable Competing interests: None
REVIEW RETURNED	09-Aug-2021

GENERAL COMMENTS	Thompson et al present a national survey of multidisciplinary PICU providers to determine current practices and perspectives of early rehabilitative therapies. Overall, the research is well-performed (esp inclusion of multiple viewpoints), and the paper is well-written. This important topic has been investigated before in Canada and in an international survey (not cited - Pediatr Crit Care Med 20(6): e274-e282); however, these data are important for regional efforts. Abstract/Methods  1. First sentence could be clarified - believe you meant to convey that most respondents were in favor of multidisciplinary involvement? Or Each discipline surveyed was interested in implementing ERM? 2. Focusing the analysis of when a child should have ERM to a cutoff of > 28 days is quite restrictive ('always' only) an interpretation. Many (?majority) respondents did seem to think 7-28 days was 'very often' reasonable. Might help your case to highlight this result in light of all of the barriers. 3. In general, helpful to state whether a result is the respondents' 'perception' or 'actual' situation. 4. If room, highlighting the # of sites that have ERM protocols up front is important to interpret rest of the results.
---

	Results  1. Helpful to have # centers with a protocol in the results, rather than discussion section. 2. Recipients of ERM. There are more children receiving ERM by % than sites that have a protocol or support the 'early' piece. 3. Initiation and delivery of ERM. Would be helpful to know how ERM is 'ordered' in UK hospitals - via electronic medical record order, etc.? Sometimes that can be a significant barrier. 4. Helpful to know (if data available) what human resources are available in the UK to provide ERM (RVU's per PICU beds?), and whether inpatient rehabilitation is available/offered post-PICU. Side note: Suggest consider how one center's experience could contribute to the overall goal of implementing ERM (includes safety and intervention guidelines): Pediatr Qual Saf 6(3): e414.
--	---

VERSION 1 – AUTHOR RESPONSE

Dear Professor Imti Choonara, MD,
The Editor-in-Chief,
BMJ Paediatrics,

We are pleased to be submitting the research report for publication in *BMJ Paediatrics: Early Mobilisation and Rehabilitation in the PICU: A UK survey*.

This is the first study to review current early mobilisation and rehabilitation practices within UK Paediatric Intensive Care Units (PICU). It has identified strong positive support for early rehabilitation and mobilisation across the NHS, but wide variation and lack of guidance in its delivery and practice. Our adult colleagues appear much further along the journey to implement this important therapy, and the COVID pandemic has significantly raised its profile in adult ICU and post-ICU practice. We are therefore working hard towards improvements for paediatric patients.

It is part of a more extensive programme of research evaluating early rehabilitation within NHS intensive care units: The Paediatric Early Rehabilitation and Mobilisation during InTensive care feasibility (PERMIT) study is funded by the National Institute of Health Research (NIHR) HTA.

We believe this study is relevant to your readership, and awareness of this emerging therapy in the UK will have a major impact on PICU and post-PICU care for generalists and specialist paediatricians.

We confirm that all authors were fully involved in the study and preparation of the manuscript and that the material within has not been and will not be submitted for publication elsewhere.

Thank you for considering our submission for publication in *BMJ Paediatrics*; we look forward to hearing from you.

Please get in touch with us if you have any questions.

Yours sincerely,

On behalf of my co-authors,
Jacqueline Y. Thompson

VERSION 2 – REVIEW

REVIEWER	Reviewer name: Dr. Sapna Kudchadkar
-----------------	-------------------------------------

	Institution and Country: United Kingdom of Great Britain and Northern Ireland Competing interests: None
REVIEW RETURNED	29-Oct-2021

GENERAL COMMENTS	The authors present findings from a survey of UK-based PICUs regarding context and professional perspectives of delivering early rehabilitation and mobilization. Topic is timely, important and directly relevant to pediatric critical care. Abstract: Results: This sections requires revision for clarity and maximizing use for the reader with quantitative context. The first section seems to be broad summary of the findings.  1. What does "a strong multidisciplinary involvement" mean? Also, "all age groups were considered"-- by whom? all respondents felt all age groups should be considered? 2. "ERM was defined by participants..."-- all participants? most? 3. "ERM was reported to be more likely to be delivered to patients when PICU length of stay exceeded 28 days"-- unclear how this fits the definition of "early" 4. Recommend ordering barriers by frequency reported, highest to lowest. 5. Recommend removing quotes in last sentence, modify to "Respondents ranked reduction in length of stay (74%) and improvement in psychological outcomes (73%) as the most important benefits of ERM" Introduction  1. Page 5/Line 1: What defines "early"? as part of the definition. 2. In the second paragraph may be worthwhile to cite recent ABCDEF bundle practice paper by Ista and mention EU findings with regards to mobility protocols. Methods:  1. Great to see patient rep was included, what about interprofessional input into the actual survey design (before pilot) Results  1. Now that I see that OTs represented 15% of responses I would include that in abstract as well and break out of "others" 2. Description of ERM-- this needs to be better summarized in the abstract with numbers per my previous point above. 3. Page 9/Lines 30-42-- There is a huge gap between 3 days and 28 days what about everyone else? This should be discussed without having to refer to the table. 4. Perceived benefits: Include the proportions after each of the outcomes identified (like done in abstract) 5. Please include "n=" in the parens when numbers are included i.e. Lines 7-12 on page 10 (n=92; 74%) 6. The free text comments seem like they could be a table on their own with themes, something to consider in lieu of pulling out two specific comments here. Discussion:  1. First sentence seems a bit broad-- did the investigators specifically query respondents about cognitive outcomes? Figure 1/survey lists psychological outcomes but only mention of "cognitive" is with regards to barriers. Would revise.
---

REVIEWER	Reviewer name: Dr. Cintia Johnston Institution and Country: University of São Paulo Institute of Biomedical Sciences, Pediatrics Competing interests: None
REVIEW RETURNED	29-Oct-2021

GENERAL COMMENTS	Dear Authors, I congratulate you for the excellent survey, very objective and clear. The writing is impeccable. The sample reached was 90% of the PICU's existing in the region where the study was carried out, which brings credibility to the results found. The theme "early mobilization and rehabilitation" in PICU still has space for surveys, so that we can understand what is happening on all continents. In this way, I believe that the study will contribute to the understanding of clinical practice and serve as a basis for strengthening early mobilization protocols (or for their development). thank You.
--

VERSION 2 – AUTHOR RESPONSE

	We would like to thank the editor and all our reviewers for the time they have taken to consider our manuscript and for giving us a chance to respond to insightful feedback. In addition to addressing the reviewers' comments we have taken this opportunity to make some minor tweaks to presentation and to correct a few minor errors which have been identified.  1. Whilst reviewing the manuscript, we noted that there was some language errors and we have taken the opportunity to correct this. Some minor formatting changes to tables have been applied to the tables on review.		
	Editor's comment	Response	
	In P5 L6, the authors state only the aims which appear broad. It would be better to state them as objectives and if possible separately for the quantitative and qualitative parts.	We have expanded the description of our aims and separated the qualitative and quantitative objectives, thank you for this comment.	New Text Our objectives were to:  I. Explore how health care professionals describe and administer ERM using a qualitative approach II. Identify and describe current ERM practice using a quantitative approach III. Understand perceived barriers and facilitators of ERM, presenting findings using descriptive analysis.
	In P6L44, provide details of response rate. It appears that the response rate was not proportional to the size of the eligible responder	Our original targeted sample were at least 3 members (Nurse, medical and physiotherapy) from each of the 29 PICUs in the UK, with an additional request for further members of the multidisciplinary team. We have therefore reported the response rate by	We received responses from PICS-SG link members in 26/29 (90%) UK PICUs. A total of 191 healthcare professionals opened the survey link, with 124 (65%) submitting responses

	categories. Was it so?	PICU in the opening paragraph of the results section. As the total sample ranged from a minimum 3x29= 87 up to 6-10 times the number of units responding, we were only able to report the responses by number of members of the multidisciplinary team who opened the survey link (eg 124/191). We hope this clarifies our sampling approach.	
	In P6L49 – The authors mention that Likert scales with median and IQR were used to report frequencies. It is not clear how this was done. Please explain.	We have provided further information on the level used to create the frequencies and percentages	Likert scales with the following categories (and associated scores) – always (5), very often (4), sometimes (3), seldom (2) and never (1) - were used to express the frequency of practice or level of agreement (using median and IQR of score).
	The authors mention a pilot phase with some changes made after the same. Please provide more details as to how changes were made after looking at the pilot data.	Apologies for the lack of clarity. We have added clarity to this stating that no questions were removed as there were few missing responses, but five questions were rephrased to add clarity	There were very few missing responses therefore no questions were removed, but five questions were rephrased to add clarity.
	The authors need to state why a validation was not attempted. This is the biggest limitation of this survey and all results and conclusion needs to be stated in this context	We did not conduct questionnaire validation because it would be very difficult to standardize these questions within the time constraints of the study and the NIHR HTA funding. This is acknowledged in the study limitations. The study team considered the questionnaire as suitable for the study aims after receiving two rounds of expert reviews.	A limitation was the use of a non-validated questionnaire
	Reviewer's comment	Response	
1	Dear Authors, I congratulate you for the excellent survey, very objective and clear. The writing is impeccable. The sample reached was 90% of the PICU's existing in the region where the study was	Thank you for your comments and time to review our manuscript.	

carried out, which brings credibility to the results found. The theme "early mobilization and rehabilitation" in PICU still has space for surveys, so that we can understand what is happening on all continents. In this way, I believe that the study will contribute to the understanding of clinical practice and serve as a basis for strengthening early mobilization protocols (or for their development). thank You.		
---	--	--

Reviewer 2

	The authors present findings from a survey of UK-based PICUs regarding context and professional perspectives of delivering early rehabilitation and mobilization. Topic is timely, important and directly relevant to pediatric critical care.		
1	Abstract: Results: This sections requires revision for clarity and maximizing use for the reader with quantitative context. The first section seems to be broad summary of the findings. What does "a strong multidisciplinary involvement" mean? Also, "all age groups were considered"-- by whom? all respondents felt all age groups should be considered?	Thank you for these helpful suggestions. We have revised this sentence .	Multi-disciplinary involvement in initiating ERM was commonly reported.

2	"ERM was defined by participants..."-- all participants? most?	Thank you for highlighting this vague sentence. We have revised this accordingly Key components of participants' definitions of ERM included tailored, multi-disciplinary rehabilitation packages, focused on promoting recovery.	Key components of participants' definitions of ERM included tailored, multi-disciplinary rehabilitation packages, focused on promoting recovery.
3	"ERM was reported to be more likely to be delivered to patients when PICU length of stay exceeded 28 days"-- unclear how this fits the definition of "early"	Thank you for highlighting this vague sentence. We have revised this accordingly	However, responses differed with regards to the timing of initiation. Interventions considered were ERM was reported to be more likely to be delivered to patients when PICU length of stay exceeded 28 days and among patients with acquired brain injury or severe developmental delay
4	Recommend ordering barriers by frequency reported, highest to lowest.	Thank you for the suggestion. We have re-ordered barrier	The most commonly identified barriers were: physiological instability (81%), limited staffing (79%), sedation requirement (73%), insufficient resources and equipment (69%), lack of recognition of patient readiness (67%), patient

		using this format	suitability (63%), inadequate training (61%), and inadequate funding (60%).
5	Recommend removing quotes in last sentence, modify to "Respondents ranked reduction in length of stay (74%) and improvement in psychological outcomes (73%) as the most important benefits of ERM"	We have removed the quotes in this sentence	Respondents ranked reduction in PICU length of stay 74% and improvement in psychological outcomes 73% as the most important benefits of ERM.
	Introduction		
1	Page 5/Line 1: What defines "early"? as part of the definition.	We have incorporated a phrase to indicate the definition of early in this sentence	Early rehabilitation and mobilisation (ERM) encompass patient-tailored interventions, delivered within seven days of admission, individually or in a bundled package to patients within intensive care settings.
2	In the second paragraph may be worthwhile to cite recent ABCDEF bundle practice paper by Ista and mention EU findings with regards to mobility protocols.	We have cited this reference and described relevant application to European PICU's	
	Methods:		
	Great to see patient rep was included, what about interprofessional input into the actual survey design (before pilot)	We did not conduct questionnaire validation because it would be very difficult to standardize these questions within the time constraints of the study and	The pilot survey received input from members of the UK Paediatric Intensive Care Society Study Group (PICS-SG) consisting of multi-disciplinary health professionals.

		the NIHR HTA funding. This is acknowledged in the study limitations. The study team considered the questionnaire as suitable for the study aims after receiving two rounds of expert reviews.	
	Results		
1	Now that I see that OTs represented 15% of responses I would include that in abstract as well and break out of "others"	We have broken the frequency to provide the response from OT's	19 (15%) occupational therapists, and 40 21 (3217%) other professionals.
2	Description of ERM-- this needs to be better summarized in the abstract with numbers per my previous point above.	We have added additional information about ERM to improve understanding	Key components of participants' definitions of ERM included tailored, multi-disciplinary rehabilitation packages, focused on promoting recovery
3	Page 9/Lines 30-42-- There is a huge gap between 3 days and 28 days what about everyone else? This should be discussed without having to refer to the table.	We have added relevant information about the other PICU days	Interventions considered were ERM was reported to be more likely to be delivered to patients when PICU length of stay exceeded 28 days and among patients with acquired

		of admission	brain injury or severe developmental delay.
4	Perceived benefits: Include the proportions after each of the outcomes identified (like done in abstract) 5. Please include "n=" in the parens when numbers are included i.e. Lines 7-12 on page 10 (n=92; 74%) 6. The free text comments seem like they could be a table on their own with themes, something to consider in lieu of pulling out two specific comments here.	Thank you for the feedback about the reporting of the free-text comments. The majority of participants completed the multiple-choice responses and did not provide additional contextual information. Where participants did, it was about a specific question; therefore, we did not feel it was appropriate to analyse thematically. We have added the quotes to add	We have included the number in parenthesis when percentages were presented

		context but are happy to remove these if the reviewers feel they are unhelpful ?	
	Discussion:		
1	First sentence seems a bit broad-- did the investigators specifically query respondents about cognitive outcomes? Figure 1/survey lists psychological outcomes but only mention of "cognitive" is with regards to barriers. Would revise.	Thank you for this comment . There was no specific question related to cognitive outcomes therefore this has been removed.	

VERSION 3 – REVIEW

REVIEWER	Reviewer name: Dr. Sapna Kudchadkar Institution and Country: United Kingdom of Great Britain and Northern Ireland Competing interests: None
REVIEW RETURNED	13-Dec-2021

GENERAL COMMENTS	Thank you for this revision in response to the reviewer comments. The manuscript is substantially strengthened by the authors. One minor point is that the Peds ABCDEF bundle manuscript and reference by Ista et al in CCM has not been incorporated as stated by the authors in the response. It appears that it has been deleted in a track changed version.
---

REVIEWER	Reviewer name: Dr. Cintia Johnston Institution and Country: University of São Paulo Institute of Biomedical Sciences, Pediatrics Competing interests: None
REVIEW RETURNED	08-Dec-2021

GENERAL COMMENTS	Dear Authors, the theme is current and of multidisciplinary interest. I consider the article well-written and that it contemplates what we proposed to accomplish in its method. Its biggest limitation is the
--

	use of an unvalidated questionnaire, a fact justified by the Authors, given the difficulties (still) of information in the pediatric population. Small suggestions: 1- update the cited references, including several publications that occurred in 2020 and 2021 2- perform agreement analysis between members from different professional areas who responded to the questionnaire (which is the agreement KAPA among the "interviewees"). This analysis, if possible, would increase the reliability of the study results/statements. Sincerely,
--	---

VERSION 3 – AUTHOR RESPONSE

Birmingham Acute Care Research Group,
 Institute of Inflammation and Ageing,
 University of Birmingham
 Birmingham,
 B15 2TT

31st January 2022

BMJ Paediatrics Open
 BMA House
 Tavistock Square
 London, WC1H 9JR
 UK

Dear Professor Imti Choonara,
 The Editor-in-Chief,
 BMJ Paediatrics Open,

We are pleased to be re-submitting the research report: Early Mobilisation and Rehabilitation in the PICU: A UK survey for publication in BMJ Paediatrics Open.

Thank you for your favourable response and suggestions to improve the manuscript.

We have addressed all the points raised by reviewers and editors with tabulated responses. In other cases, we explained when the authors implemented no changes. Thank you for considering our submission for publication; we look forward to hearing from you.

Yours sincerely,

On behalf of my co-authors,
 Jacqueline Y. Thompson

VERSION 4 – REVIEW

REVIEWER	Reviewer name: Dr. Cintia Johnston Institution and Country: University of São Paulo Institute of Biomedical Sciences, Pediatrics Competing interests: None
REVIEW RETURNED	04-Feb-2022
GENERAL COMMENTS	The authors carried out all the suggested corrections.

REVIEWER	Reviewer name: Dr. Sapna Kudchadkar Institution and Country: United Kingdom of Great Britain and Northern Ireland Competing interests: None
REVIEW RETURNED	06-Feb-2022

GENERAL COMMENTS	I think there is confusion about the Ista citation that was recommended to be added. First, reference #12, while it should absolutely be included, is incorrect. It appears to be a reference to an abstract but should instead be: https://pubmed.ncbi.nlm.nih.gov/32576273/ In addition to the point prevalence study in EU this reviewer was referring to https://pubmed.ncbi.nlm.nih.gov/34259659/ to put EU/UK practice in context of international practice.
--

VERSION 4 – AUTHOR RESPONSE

Birmingham Acute Care Research Group,
Institute of Inflammation and Ageing,
University of Birmingham
Birmingham,
B15 2TT

7th February 2022

BMJ Paediatrics Open
BMA House
Tavistock Square
London, WC1H 9JR
UK

Dear Professor Imti Choonara,
The Editor-in-Chief,
BMJ Paediatrics Open,

We are pleased to be re-submitting the research report: Early Mobilisation and Rehabilitation in the PICU: A UK survey for publication in BMJ Paediatrics Open.

We are pleased to be re-submitting the research report: Early Mobilisation and Rehabilitation in the PICU: A UK survey for publication in BMJ Paediatrics Open.

Thank you for your favourable response and for the suggestions to improve the manuscript.

We have addressed all the points about references raised by reviewers and editors with tabulated responses. Thank you for considering our submission for publication; we look forward to hearing from you.

Yours sincerely,

On behalf of my co-authors,
Jacqueline Y. Thompson

VERSION 5 – REVIEW

REVIEWER	Reviewer name: Dr. Sapna Kudchadkar Institution and Country: United Kingdom of Great Britain and Northern Ireland
-----------------	--

	Competing interests: None
REVIEW RETURNED	24-Feb-2022

GENERAL COMMENTS	All revisions have been made and this is an outstanding addition to the literature.
---